# Contractile Work of the Soleus and Biarticular Mechanisms of the Gastrocnemii Muscles Increase the Net Ankle Mechanical Work at High Walking Speeds

**DOI:** 10.3390/biology12060872

**Published:** 2023-06-16

**Authors:** Mohamadreza Kharazi, Christos Theodorakis, Falk Mersmann, Sebastian Bohm, Adamantios Arampatzis

**Affiliations:** 1Department of Training and Movement Sciences, Humboldt-Universität zu Berlin, 10115 Berlin, Germany; mohamadreza.kharazi@hu-berlin.de (M.K.); theodchr@hu-berlin.de (C.T.); falk.mersmann@hu-berlin.de (F.M.); sebastian.bohm@hu-berlin.de (S.B.); 2Berlin School of Movement Science, 10115 Berlin, Germany

**Keywords:** energy transfer, ankle power, Achilles tendon force, energy storage and recoil, maximum walking speed

## Abstract

**Simple Summary:**

This study enhances our understanding of how different mechanisms in the triceps surae muscles contribute to increased mechanical power and work at the ankle joint during various walking speeds. The findings reveal that both the soleus (Sol) and gastrocnemii muscles play a role in the speed-related increase of mechanical work at the ankle joint but through distinct energetic processes and mechanisms. The Sol muscle primarily increases contractile work (62%), while the gastrocnemii muscles demonstrate an enhanced contribution through biarticular mechanisms (38%). This knowledge can inform the design of exercise interventions and customized assistance through bioinspired exoskeleton configurations for both healthy individuals and those with pathological conditions.

**Abstract:**

Increasing walking speed is accompanied by an increase of the mechanical power and work performed at the ankle joint despite the decrease of the intrinsic muscle force potential of the soleus (Sol) and gastrocnemius medialis (GM) muscles. In the present study, we measured Achilles tendon (AT) elongation and, based on an experimentally determined AT force–elongation relationship, quantified AT force at four walking speeds (slow 0.7 m.s${}^{-1}$, preferred 1.4 m.s${}^{-1}$, transition 2.0 m.s${}^{-1}$, and maximum 2.6 ± 0.3 m.s${}^{-1}$). Further, we investigated the mechanical power and work of the AT force at the ankle joint and, separately, the mechanical power and work of the monoarticular Sol at the ankle joint and the biarticular gastrocnemii at the ankle and knee joints. We found a 21% decrease in maximum AT force at the two higher speeds compared to the preferred; however, the net work of the AT force at the ankle joint (ATF work) increased as a function of walking speed. An earlier plantar flexion accompanied by an increased electromyographic activity of the Sol and GM muscles and a knee-to-ankle joint energy transfer via the biarticular gastrocnemii increased the net ATF mechanical work by 1.7 and 2.4-fold in the transition and maximum walking speed, respectively. Our findings provide first-time evidence for a different mechanistic participation of the monoarticular Sol muscle (i.e., increased contractile net work carried out) and the biarticular gastrocnemii (i.e., increased contribution of biarticular mechanisms) to the speed-related increase of net ATF work.

## 1. Introduction

Human walking covers a wide range of speeds, from 0.4 to 3.0 m.s${}^{-1}$ [1,2,3]. During walking, the mechanical work performed at the ankle joint accounts for 40 to 50% of the total mechanical work carried out in the lower extremities [4,5]. Furthermore, increasing walking speed is accompanied by an increase of the mechanical power and work at the ankle joint [5,6]. These reports give evidence for the relevant contribution of the ankle joint to the necessary mechanical power and work during walking and thus indicate the important role of the plantar flexor muscles as a key source of energy production for walking. The monoarticular soleus (Sol), as the most voluminous muscle of the plantar flexors, generates the most significant part of the mechanical power and work at the ankle joint during walking [7]. The biarticular gastrocnemius medialis (GM) and lateralis (GL) muscles generate moments in both the ankle and knee joints. Thus, power and energy can be transferred from the more proximal monoarticular vasti muscles to the ankle joint [8,9,10]. With this, biarticular muscles may regulate the redistribution and transfer of mechanical power and energy between the crossing joints to be effective at the joint where required [11,12]. Observations in turkeys, wallabies, and goats showed that biarticular muscles primarily transfer energy between two crossing joints, significantly contributing to increased positive work during level locomotion [13]. In running dogs, the contribution of the knee-to-ankle joint energy transfer was reported to be around 49% of the total positive work performed at the ankle joint [14]. Two separate mechanisms of the biarticular gastrocnemii muscles can influence the mechanical power and work at the ankle joint, independent of their musculotendinous power and work production [10,15,16]. First, an energy transfer between the two joints is possible when the mechanical powers of the gastrocnemii muscles at the ankle and knee joint have opposite signs (energy transfer mechanism). Second, the gastrocnemii can simultaneously absorb (negative) or generate (positive) mechanical power and work at the two crossed joints, thus affecting the redistribution of power and work between the two joints (joint coupling mechanism). The function of the biarticular gastrocnemii muscles at the ankle and knee joints during high walking speeds and the possible modulation of biarticular mechanisms with increasing walking speeds to enhance power and work production at the ankle joint is currently unknown.

It is well known that the energy expenditure per meter distance or the metabolic energy cost of walking as a function of walking speed is characterized by a U-shaped curve, with a minimum at speeds around 1.4 m.s${}^{-1}$ [17,18,19,20,21]. Humans’ preferred walking speed is very close to the optimum speed that minimizes the metabolic energy cost of transport [3,19]. It is also accepted that humans switch voluntarily from walking to running at speeds around 2.0 m.s${}^{-1}$, although the metabolic energy cost of running is higher than that of walking [5,22,23]. The higher metabolic cost of running at 2.0 m.s${}^{-1}$ shows that the transition from walking to running is not primarily triggered to minimize the energy costs of locomotion. Noble et al. [24] found that the perception of exertion is lower in a wide range of speeds while walking than during running, yet in the walking-to-running transition speed, the perception of exertion during walking exceeded running and, as a result, humans decided to switch from walking to running. Changes in the activation of the lower extremity muscles may increase the sensation of effort and trigger the transition from walking to running [25,26]. A recent study [27] reported that locomotor control and decision making for a less overburdening walking condition are regulated not only based on the whole-body metabolic cost of walking but also muscle activation with the objective to minimize local muscle fatigue. Minetti et al. [23] suggested that the plantar flexor muscles begin to work inefficiently due to the high contraction velocities at the walking-to-running transition speed. Therefore, humans switch to running to improve the efficiency of the plantar flexor muscles, despite the higher metabolic energy cost. Switching from walking to running at transition speed resulted in a greater force–velocity potential (i.e., decreased fascicle shortening velocity) for the GM muscle [28] and greater force–length–velocity potential (i.e., operating fascicle length close to the optimal length and decrease fascicle shortening velocity) for the Sol muscle [6], which in turn increases the economy of muscle force generation [29]. These results further support model predictions from Neptune and Sasaki [30] that intrinsic contractile properties of the plantar flexor muscles might be responsible for the transition from walking to running.

Nevertheless, humans can walk at higher speeds than the transition speed, despite the above-mentioned decreased potential of the Sol and GM muscles to generate forces [1,2,25]. The impaired muscle force potential at walking speeds of 2.0 m.s${}^{-1}$ [6,28] may suggest lower generated muscle forces at higher walking speeds. Inverse dynamic approaches, however, report greater resultant ankle joint moments in walking speeds around the transition speed (2.0 to 2.1 m.s${}^{-1}$) compared to the preferred speed [6,31], indicating an increased muscle force generation of the plantar flexor muscles. The ankle joint moments calculated using inverse dynamics are the result of all forces acting around the ankle joint (i.e., forces of synergist and antagonist muscles, forces transmitted by ligaments, bone-to-bone contact forces, and forces from soft tissues around the ankle joint). Therefore, the calculated maximum ankle joint moments may not represent the moments generated by the plantar flexor muscles at the ankle joint. Using the optic fiber methodology, Finni et al. [32] measured similar peak forces in the Achilles tendon (AT) at walking speeds between 1.1 and 1.8 m.s${}^{-1}$, indicating an unchanged muscle force generation in this range of speeds. Finni et al. [32] did not measure AT forces in walking speeds at ≥2.0 m.s${}^{-1}$, thus possible modifications in the AT force cannot be excluded. To our knowledge, measurements of the AT forces at walking speeds higher than the transition speed have not been conducted yet. Assuming a lower muscle-force generation in the transition and higher walking speeds, and thus lower energy storage and recoil from the AT compared to the preferred speed, we can expect compensatory mechanisms of power and energy production to provide the necessary mechanical power and work at the ankle joint. Using a musculoskeletal model, Neptune et al. [7] found that the monoarticular Sol and the biarticular gastrocnemii increase the net contractile work in the transition speed compared to those close to the preferred one. To our knowledge, this model prediction has not yet been experimentally validated. Furthermore, there is a lack of information concerning the AT elastic energy storage and recoil and the musculotendinous power and work performed by the main plantar flexor muscles at maximum walking speeds.

The purpose of the current study was to investigate the contribution of monoarticular and biarticular mechanisms of the Sol and gastrocnemii muscles to the mechanical power and work at the ankle joint from slow to maximum walking speeds. This knowledge may be useful for designing exercise interventions and bioinspired exoskeleton configurations for customized assistance in healthy and pathological conditions. We used an innovative approach [33] to measure the AT elongation during walking (Figure 1) and, based on an experimentally determined tendon force–elongation relationship as a calibration measure, we quantified the AT force in four walking speeds (slow 0.7 m.s${}^{-1}$, preferred 1.4 m.s${}^{-1}$, transition 2.0 m.s${}^{-1}$, and maximum 2.6 ± 0.3 m.s${}^{-1}$). Further, we measured the electromyographic activity (EMG) of the Sol, GM, and tibialis anterior (TA) muscles, and the AT elastic strain energy storage and recoil were calculated. Finally, we investigated the mechanical power and work of the AT force at the ankle joint and, separately, the mechanical power and work of the monoarticular Sol at the ankle joint and the biarticular gastrocnemii muscles at the ankle and knee joints. We hypothesized that an increase in contractile net work production of the Sol muscle and an increased contribution of biarticular mechanisms of the gastrocnemii muscles would result in a continuous increase of net mechanical work at the ankle joint from slow to maximum walking speed.

## 2. Materials and Methods

### 2.1. Experimental Design

Fifteen healthy individuals (four female) with no record of musculoskeletal disorders (age 28 ± 4 years, height 175.0 ± 7.5 cm, body mass 75.0 ± 9.5 kg) participated in the study. All participants submitted written informed consent to the experimental procedure that was approved by the ethics committee of the Humboldt-Universität zu Berlin (HU-KSBF-EK-2018-0005). The participants were instructed to walk at the speeds of 0.7 m.s${}^{-1}$ (slow walking), 1.4 m.s${}^{-1}$ (preferred walking), 2.0 m.s${}^{-1}$ (transition walking), and their maximum walking speed capacity (2.6 ± 0.3 m.s${}^{-1}$) on a treadmill (Daum Electronic, Ergorun Premium8, Fürth, Germany). Experimentally determined individual preferred walking speeds in previous studies ranged between 1.1 and 1.6 m.s${}^{-1}$ [19,34,35], and the walking-to-running transition speed between 1.88 and 2.20 m.s${}^{-1}$ [21,22,36]. Here we used average values of 1.4 and 2.0 m.s${}^{-1}$, similar to other studies [6,28], as the main purpose was not to explain why humans decide to walk at their preferred speed or transition from walking to running but to investigate the contribution of monoarticular and biarticular mechanisms to the mechanical power and work at the ankle joint with increasing walking speed. This means that in our study, we investigated the average preferred and transition walking speeds in the included participants. Individual maximum walking speed capacity was determined by gradually increasing the treadmill speed, starting from the transition speed (i.e., 2.0 m.s${}^{-1}$), while the participants were instructed to maintain their walking pattern. As soon as the walking pattern could not be maintained and a flight phase was introduced, the earlier selected speed was set as the maximum walking speed.

### 2.2. Kinematics and Electromyographic Activity Measurements

Six reflective markers (14 mm in diameter) were placed on anatomical landmarks, i.e., the tip of the second metatarsal, medial, and lateral epicondyles of the femur, midpoint of a straight line between the greater trochanter and the lateral epicondyle of the femur, and medial and lateral malleolus. One additional reflective foil marker (5 mm in diameter, flat surface) was placed on the insertion of the AT on top of the calcaneus (heel marker, Figure 1). The insertion point of the AT was defined as the notch of the calcaneus bone, which was determined by moving a sound absorptive marker underneath an ultrasound probe in the sagittal plane until the shadow reflection (i.e., caused by sound absorptive marker) crossed the notch. Fourteen Vicon (Version 1.8.1, Vicon Motion Systems, Oxford, UK, 4 MX T20, 2 MX-T20-S, 6 MX F20, 2 MX F40, 250 Hz) cameras were used to capture the 3D trajectories of all markers in real-time. A fourth-order, low-pass, zero-phase shift Butterworth filter with a cut-off frequency of 12 Hz was applied to the raw marker trajectories. The participants performed a one-minute familiarization on the treadmill for each speed. The touchdown of the foot was determined as the heel marker’s instant minimal vertical position [37]. The foot take-off was defined as the sign change in the second metatarsal marker velocity in the anterior-posterior direction [38,39]. The ankle joint angle in the sagittal plane was calculated as the angle between the foot (i.e., line crossing the tip of the second metatarsal and calcaneus marker) and shank (i.e., line crossing ankle joint center and knee joint center). Ankle and knee joint centers were defined as the mean point between lateral and medial reflective markers on the malleoli and femur condyles. The knee joint angle in the sagittal plane was calculated as the angle between the femur axis (i.e., the line crossing the lateral knee and greater trochanter reflective marker) and the shank. The ankle and knee joint angles were calculated referencing a neutral quiet stance position (the foot perpendicular to the tibia represents a 0° ankle joint angle; the knee fully extended represents a 0° knee joint angle). Positive values of ankle and knee joint angles constitute ankle plantar flexion and knee flexion. The stance time was calculated as the time duration from the right touchdown and take-off. Accordingly, swing time was calculated from take-off to the next touchdown. Cadence was calculated as a total number of touchdowns divided by the two-fold total time of nine gait cycles. Step length was calculated as the treadmill speed divided by cadence. The duty factor was calculated as the stance time divided by the sum of stance and swing time. A wireless system (Myon m 320RX, Myon AG, Baar, Switzerland) was used to measure the surface EMG activity of the TA, GM and Sol muscles during walking, operating at a sampling frequency of 1000 Hz. A fourth-order high-pass zero-phase shift Butterworth filter with a 50 Hz cut-off frequency, a full-wave rectification, and a low-pass zero-phase shift filter with a 20 Hz cut-off frequency was used to process the raw EMG signal. The resulting EMG signal was normalized to the highest processed EMG value acquired during a maximum voluntary isometric contraction (MVC).

### 2.3. Measurement of AT Length and Quantification of AT Force and Strain Energy

The origin of the AT was determined as the most distal junction between the AT and the medial head of the gastrocnemius muscle (GM-MTJ), obtained by transverse and sagittal ultrasound scans (Figure 1). A 60 mm T-shape ultrasound probe (Aloka UST-5713T, Hitachi Prosound, alpha 7, Hitachi, Japan) with a sampling frequency of 146 Hz was fixed above the GM-MTJ and was tightened firmly with a flexible plastic cast. An ultrasound gel pad was placed between the ultrasound probe and the skin to compensate for the unevenness of the skin surface. The ultrasound and motion-capture systems were synchronized using a 5-volt manual trigger through the ultrasound electrocardiograph channel and motion-capture analog input channel. A semi-automatic image-based tracking algorithm was implemented in a self-developed user interface (MATLAB, version 9.6. Natick, MA, USA: The MathWorks Inc.) to track the position of the GM-MTJ from the stack of ultrasound images. The details of the developed algorithm and its validity compared with manual tracking during human walking and running (i.e., $r^{2}$ = 0.97) were provided earlier [33]. The skin surface was detected using a double threshold-to-intensity gradient provided by the Canny edge-detection function [40]. A third-order polynomial curve was then fitted to the detected points on the skin, and the tracked coordinates of the GM-MTJ were perpendicularly (i.e., shortest distance) projected to the fitted curve.

For the transformation of the GM-MTJ to the global motion-capture coordinate system, the four corners of the ultrasound probe’s protective plastic rubber were digitized in 3D space using a custom 3D-printed calibration tool in a separate session. A coordinate system was defined on the left-center side of the plastic rubber. The gap between the origin of the specified coordinate system and the first piezoelectric crystal underneath was determined by subtracting the ultrasound rubber plastic width from the ultrasound image width. A custom 3D-printed tripod was mounted on the ultrasound probe, and a coordinate system was defined accordingly. The projected positions of the GM-MTJ on the skin surface were then transferred to the global motion capture coordinate system (for details, see [33]). Depending on the location of the GM-MTJ, reflective foil markers were placed on the skin in 20 mm intervals from the calcaneus bone (i.e., insertion point of the AT, described above) on the path of the AT to the last possible position below the cast (Figure 1). The curved length of the AT was determined as the sum of vectors using the 3D coordinates of every two consecutive foil markers and the GM-MTJ projected to the skin. The strain of the AT was calculated by dividing the measured AT elongation by the AT length measured at a relaxed state in 20° plantar flexion, where AT slackness has been reported [41]. Potential displacements of the skin to the bone underneath the calcaneus marker that defines the AT insertion can also introduce artifacts in the AT length measurement [42]. In a separate experiment, kinematic and ultrasound measurements were used to measure the displacement of the reflective calcaneus marker above the defined insertion point (notch at tuber calcanei) to account for the potential movements of the skin relative to the calcaneus bone [33]. For this purpose, a sound-absorptive marker was placed between the ultrasound probe and skin close to the bony insertion, and the dynamometer (Biodex Medical, Syst.3, Shirley, NY, USA) passively rotated the ankle joint. The displacement of the calcaneus bone relative to the skin as a function of the heel angle (angle between the line from the calcaneus marker to ankle joint center and the line from the ankle joint center to the knee joint center) was determined and used as a model to correct the AT length during walking as a function of the heel angle. The potential error due to the skin–bone artifact was 0.92 ± 0.57 mm (mean ± standard deviation) on the AT length and 0.45% on AT strain [33].

In an additional session, an individual force–elongation relationship of the AT was determined. For the measurement of the plantar flexion moment, five ramp MVC (5 s gradual increase of the moment exertion) plantar flexions were performed at a joint angle position of 0° and fully extended knee on a dynamometer. The potential misalignments of ankle and dynamometer axes of rotation during the contractions and gravitational and passive moments were considered by employing an inverse dynamics approach [43]. Further, we considered the effect of coactivation of the antagonistic muscle (i.e., tibialis anterior) on the resultant joint moment during the MVCs using an established EMG-based approach [44]. The ankle joint moment was then divided by the AT lever arm to calculate the tendon force. For this purpose, the AT lever arm was calculated using the tendon excursion method and adjusted for the changes in the alignment of the AT during the contractions using the data provided by Maganaris et al. [45]. A 10-centimeter linear ultrasound probe (My lab60, Esaote, Genova, Italy, 25 Hz) was fixed above the GM-MTJ to measure the corresponding elongation of the AT during the five MVCs. The GM-MTJ displacement resulting from ankle joint rotations (i.e., during MVC) was considered by tracking its displacement during a passive rotation (at 5°/s) of the ankle joint [46]. To achieve excellent reliability, the tendon force-elongation relationship of each participant was averaged from five MVCs [47]. The individual force-elongation relationship of the AT was obtained by fitting a quadratic function (Equation (1)), which was then used to assess the AT force during walking.
(1)$$F=a\times l^{2}+b\times l$$
where *F* is the AT force, *a* and *b* are the coefficients of the quadratic function, and *l* is the elongation of AT during the MVCs. Using Equation (1) and the measured AT elongation during the walking trials, we calculated the AT force and then AT elastic strain energy as the integral of the AT force over AT elongation (Equation (2)) during the stance phase.
(2)$$E=\int F.dl=\int \left(a\times l^{2}+b\times l\right).dl=\frac{1}{3}a.l^{3}+\frac{1}{2}b.l^{2}+c$$
where the *E* and *F* are the AT strain energy and force. *l* is the instantaneous elongation of AT during stance, and *c* is the constant of the integral. The AT power was calculated as the first time-derivative of the AT elastic strain energy. It is to be mentioned that Equations (1) and (2) do not consider the effect of tendon hysteresis on the assessed AT force and AT strain energy. However, in our earlier study [33], we estimated a negligible effect of 46 Newton and 0.2 Joule in forces and energy during locomotion, respectively (i.e., 1.9% of the maximum AT force or strain energy).

### 2.4. Mechanical Power and Work at the Ankle and Knee Joint

The ankle joint moment resulting from the AT force (ATF moment) during walking was calculated as the product of the AT force and the AT lever arm. The instant AT lever arm was measured as the perpendicular distance between the AT line of action and the marker-based ankle joint center during the walking trials (Figure 1). The line of action of the AT was assumed at the midpoint of the AT. The distance between the skin surface and the midpoint of the AT was assessed using sagittal plane ultrasound scans and subtracted from the kinematic-based AT lever arm. The ultrasound probe was placed above the AT on the level of the malleoli while participants were seated in a relaxed prone position with their ankle angle set at 0°. The depth between the skin surface and the midpoint of the AT among all individuals was, on average, 5.0 ± 0.3 mm. The mechanical power at the ankle joint from the AT force (ATF power) was calculated as the product of ATF moment and ankle angular velocity. The mechanical work done from the AT force (ATF work) at the ankle joint was calculated as the integral of ATF power over time.

The Sol, GM, and GL muscle forces have been assessed as a fraction of the AT force by using the relative physiological cross-sectional area of the three muscles (i.e., 62% for the Sol, 26 % for the GM, and 12% for the GL) as reported by Albracht et al. [48], assuming the force contribution of each muscle is proportional to its physiological cross-sectional area. The moment generated by the Sol muscle at the ankle joint was calculated as the product of the Sol muscle force (equal to 0.62 of the AT force) and the instantaneous AT lever arm measured during walking. The mechanical power of the Sol at the ankle joint was calculated as the product of the Sol ankle joint moment and ankle joint angular velocity. This Sol mechanical power at the ankle joint is equal to the mechanical power of the Sol muscle-tendon unit (MTU) calculated as the product of the Sol force and Sol MTU velocity [15]. The ankle mechanical work performed by the Sol was then calculated as the integral of the Sol ankle joint power over time. The GM and GL, as biarticular muscles, can produce or absorb power and energy in both the ankle and knee joints. This means they can redistribute the mechanical power and work performed by their MTUs over the two joints and transfer power and energy from knee-to-ankle joint and vice versa. We investigated the mechanical power and work at the ankle and knee joints by the GM and GL muscles in order to examine their contribution to ankle and knee mechanical power and work and the energy transfer between the two joints during walking. The mechanical power of the GM and GL muscles at the ankle/knee joints was calculated as the product of the ankle/knee joint moment generated from the two muscles and ankle/knee joint angular velocity. The generated moment at the ankle/knee joint by the two muscles was calculated as the product of the GM and GL muscle forces and their instantaneous lever arms. For the ankle joint, the measured AT lever arm was used. The lever arm of the GM and GL at the knee joint were extracted as a function of the knee joint angle using the values reported by Buford et al. [49]. The mechanical work of the GM and GL muscles at the ankle and knee joints was then calculated as the integral of the GM/GL ankle and knee joint power over time. The sum of the mechanical power/work at the ankle and knee joints of the GM and GL muscles equals the mechanical power/work of the GM/GL MTUs calculated as the product of their forces and MTU velocities [15]. In the analysis, the sum of the power and work of the GM and GL was used and all muscle kinetics data will be presented as values of the gastrocnemii muscles.

### 2.5. Statistics

A linear mixed model was used to test for the main effect of walking speed on all investigated outcomes (i.e., spatiotemporal gait parameters, kinematics, EMG, strain, moments, force, and mechanical work). In the linear mixed-effects model, the participants were treated as effects and walking speed as a fixed effect. Linear mixed models are robust against violations of the normality assumption [50], which was not given for the AT force, AT strain energy, and ATF work, according to the Shapiro–Wilk test applied to the normalized residuals. The significance level was set to α = 0.05, and all values are reported as mean ± standard errors. A pairwise Tukey test was performed as a post hoc analysis in case of a significant main effect of speed, and Benjamini–Hochberg-corrected p-values will be reported. The statistical analyses were conducted using R v4.0.1 (R foundation for statistical computing, Vienna, Austria. Packages), where the “nlme” package was used for the linear mixed model and the “emmeans” package for post hoc testing.

## 3. Results

We found a significant main effect of walking speed on all spatiotemporal gait parameters (Table 1, *p* < 0.001). Both stance and swing time were reduced significantly (*p* < 0.001) at a higher walking speed. Step length and cadence increased as a function of speed (*p* < 0.001), and the duty-factor decreased significantly (*p* < 0.001) with increasing speed; however, it remained above 0.5 (i.e., the threshold for walking) at all walking speeds (Table 1). Although the time of the dorsiflexion phase decreased significantly (*p* < 0.001) as a function of speed, the plantar flexion time remained quite preserved (Table 1). Ankle and knee joint angles (Figure 2B,C), strain (Figure 2A), force (Figure 2D), and lever arm (Figure 2E) of the AT and ankle joint moments generated by the AT force (Figure 2F) during the stance phase at all walking speeds are shown in Figure 2. There was a significant main effect of walking speed on maximum AT strain (*p* = 0.014; Table 2) and AT force (*p* = 0.005; Table 2). The post hoc comparisons showed a lower maximum AT force in the transition (*p* = 0.004) and maximum walking speed (*p* = 0.047) compared to the preferred one. The maximum AT strain was lower in the transition compared to the preferred speed (*p* = 0.017, Table 2) and showed a tendency (*p* = 0.058) towards lower values in the maximum walking speed compared to the preferred one. We found a main effect of speed on the average AT lever arm (*p* = 0.003). The AT lever arm in the transition speed was longer than in the slow (*p* = 0.002) and maximum (*p* = 0.020) walking speeds (Figure 2E). However, the differences in the AT lever arms were rather small, with 1 mm between the investigated walking speeds (i.e., 2% of AT lever arm, Table 2).

There was a significant main effect of walking speed (*p* < 0.001) on the maximum ATF moment (Table 2). Post hoc comparisons demonstrated a lower maximum ATF moment in the transition compared to the preferred (*p* = 0.005, Table 2) speed and a tendency for lower values in maximum walking speed (*p* = 0.050). In the slow and preferred speed, the initiation of the ankle plantar flexion and knee flexion in the second part of the stance phase occurred almost simultaneously (Figure 2B,C). In the two higher walking speeds, the time-wise earlier initiation of the plantar flexion was combined with a knee extension (Figure 2B,C). There was a significant effect of speed (*p* < 0.001) on the range of motion (RoM) at the ankle joint in both the dorsiflexion and plantar flexion phase (Table 2). The RoM during dorsiflexion was the lowest in the transition speed, whereas the RoM during plantar flexion was the highest at the maximum speed (Table 2). During the simultaneous plantar flexion and knee extension, the knee joint extended by 7.5 ± 1.2° in the transition and by 8.7 ± 1.3° in the maximum walking speeds. The mechanical power (Figure 3A–D) and work (Figure 3E–H) at the ankle joint from the AT force (ATF power/work) and EMG-activity of Sol, GM, and TA muscles (Figure 3I–L) are presented in Figure 3. Speed affected the ATF work during both dorsiflexion and plantar flexion periods (*p* < 0.001, Table 3). The ATF work during the dorsiflexion was the lowest in the transition speed, and during the plantar flexion, the highest in the maximum speed (*p* < 0.001, Figure 3F,H and Table 3). The net ATF work demonstrated a significant main effect of speed (*p* < 0.001) with a continuous increase as a function of walking speed (Figure 3E–H and Table 3).

The maximum AT strain energy storage and recoil was also affected by walking speed (*p* < 0.05). The energy storage and recoil were significantly lower in the transition speed (*p* = 0.011) and showed a tendency for lower values in the maximum speed (*p* = 0.084) compared to the preferred one (Table 3). The AT strain energy recoil was significantly higher than the net ATF work in the slow (*p* < 0.001) and preferred (*p* = 0.01) speed, while in the two higher speeds, the net ATF work was 1.6 to 1.9-fold greater than the AT energy recoil. The maximum EMG-activity of all three investigated muscles demonstrated a significant speed effect (*p* < 0.001) with the highest values in the transition and maximum speeds (*p* < 0.001; Table 3).

Figure 4 shows the mechanical power and work at the ankle joint by the Sol muscle, the mechanical power and work at the ankle and knee joints by the biarticular gastrocnemii muscles, and the total mechanical power and work of the gastrocnemii MTU. The Sol net work increased continuously (*p* < 0.001, Table 4, Figure 4E–H) with walking speed. The positive work of the Sol muscle was the highest at the maximum speed, whereas at the transition speed, the negative Sol work was the lowest (*p* < 0.001, Table 4). The increased net Sol work at the preferred compared to the slow speed occurred due to an average decrease of negative work and increase of positive work in the preferred speed, though the differences did not reach statistical significance (Table 4). The negative and positive work at the ankle and knee joints by the gastrocnemii muscles showed a significant speed effect (*p* < 0.001, Table 4). The negative work at the ankle joint was the lowest in the transition speed, and the positive work was the highest in the maximum walking speed (*p* < 0.001, Table 4). Between the other speeds, neither the negative nor the positive work showed any statistically significant (*p* > 0.05) differences (Table 4). The net mechanical work at the ankle joint from the gastrocnemii muscles increased significantly (*p* < 0.001) from the slow to maximum walking speed (Table 4). At the knee joint, the negative work increased, and the positive work decreased by increasing walking speed (*p* < 0.05, Table 4). As a result, the net mechanical work of the gastrocnemii at the knee joint increased in negative values as a function of speed (*p* < 0.001). The negative (*p* = 0.098), positive (*p* = 0.149), and net (*p* = 0.323) mechanical work of the gastrocnemii MTU did not show a significant speed effect (Table 4).

There were four characteristic phases during the stance phase of walking where the unique property of the biarticular gastrocnemii muscles to act simultaneously at the ankle and knee joints influenced the net mechanical work at the ankle joint despite the unchanged net mechanical work of their MTU (Figure 2B,C and Figure 4I–L). First, at the beginning of the stance phase and during the knee flexion, power and energy were transferred from the dorsiflexed ankle joint to the knee joint (phase T1, energy-transfer mechanism). Second, in the phase where the ankle joint continued the dorsiflexion and the knee joint was extended, the gastrocnemii absorbed energy in both the ankle and knee joint (phase C1, joint-coupling mechanism). Third, during a simultaneous plantar flexion and knee extension, particularly in the transition and maximum walking speeds, energy was transferred from the knee to the ankle joint (phase T2, energy-transfer mechanism). Fourth, at the end of the stance phase and during the synchronous plantar flexion and knee flexion, the gastrocnemii generated work in both the ankle and knee joint (phase C2, joint-coupling mechanism). Figure 5A visualizes the four phases, showing the mechanical power performed by the biarticular gastrocnemii muscles at the knee joint during the maximum walking speed. The mechanical knee joint work of the gastrocnemii muscles in the four phases and their net knee work during all walking speeds are shown in Figure 5B.

## 4. Discussion

Our results show that an increase in walking speed is associated with increased net ATF mechanical work, despite a decrease in the maximum AT force in the transition and maximum compared to the preferred speed. We demonstrated that the enhanced musculotendinous energy production of the monoarticular Sol and biarticular mechanisms of the gastrocnemii muscles contributed to the speed-related increase of net ATF mechanical work. Furthermore, our findings show that in the two higher walking speeds, an earlier plantar flexion due to a rapid increase in the EMG-activity of the Sol and GM muscles was accompanied by a knee extension. The consequence was a knee-to-ankle joint energy transfer via the biarticular gastrocnemii muscles and earlier mechanical energy production of the Sol MTU compared to the slow and preferred speeds. The continued elongation of the AT in the two higher speeds shows that a part of this energy was transferred to the tendon despite the initiation of the plantar flexion. In line with our hypothesis, the results show a different mechanistic participation of the monoarticular Sol muscle and the biarticular gastrocnemii in the speed-related increase of net ATF mechanical work.

Similarly to earlier studies [7], we found a continuous decrease in stance and swing times and an increase in step length and cadence as a function of walking speed. The duty factor in all investigated speeds was >0.5, showing a double contact time at all speeds. The maximum AT forces ranged between 1.616 and 2.174 kN and were very close to the measured values in comparable walking speeds reported by Finni et al. [32] using optic fiber methodology. The maximum AT force reduced by 21% in the two higher speeds compared to the preferred one, despite the higher EMG activity of the Sol and GM muscles, which confirms the predictions from musculoskeletal models at comparable walking speeds [30]. The increase of net ATF work with speed can result from both an increased net mechanical work of the contractile elements of the Sol, GM, and GL muscles and a modulation of the biarticular mechanisms. The most voluminous monoarticular Sol muscle can produce power and work only at the ankle joint. Based on the physiological cross-sectional area within the Sol, GM, and GL muscles [48], we assumed a 38% contribution of the gastrocnemii to the AT force. This means that most of the speed-related increased net ATF work is done by the Sol muscle (i.e., 62%) or, more specifically, by the Sol contractile element, as the elastic element cannot produce positive net work. The unchanged net mechanical work of the gastrocnemii MTU indicates a similar net mechanical work of their contractile elements within the four investigated speeds. This implies that the contractile elements of the gastrocnemii muscles did not contribute to the speed-related increased net ATF work.

The two gastrocnemii are biarticular muscles and, thus, are able to generate power and work in both the ankle and knee joints. Due to their biarticularity, two additional mechanisms can affect the net ATF work in addition to the work output of their contractile elements. The net mechanical work of the gastrocnemii muscles at the ankle joint showed a continuous increase from slow to the maximum speed of 4.3 ± 0.8 J, demonstrating relevant participation of the biarticular mechanisms. This speed-related increase of the biarticular net work at the ankle joint without changes in the net work of the gastrocnemii MTU was due to the speed-related increase negative net mechanical work of the gastrocnemii at the knee joint. These findings provide first-time evidence that the biarticular gastrocnemii muscles modulate their energy output within the two joints towards an increased output at the ankle joint at higher walking speeds. Thus, the current study demonstrates that both the monoarticular Sol and the biarticular gastrocnemii contribute to the speed-related increase of net mechanical work carried out at the ankle joint. However, the energetic processes and the mechanisms for this phenomenon differ between the two synergistic muscles. The Sol muscle increased the net mechanical work at the ankle joint by a speed-related increase of contractile work production (62% contribution), while the gastrocnemii muscles increased their work output at the ankle joint by an increased contribution of biarticular mechanisms (38%). The largest energy transfer from the knee to the ankle joint occurred at the maximum speed. More than three decades ago, van Ingen Schenau [12] mentioned that the timing of the coupling between knee extension and plantar flexion could influence the effectivity of the knee-to-ankle joint energy transfer via the biarticular gastrocnemii muscles. The greater knee-to-ankle joint energy transfer in the maximum walking speed indicates a more effective coupling between knee extension and plantar flexion concerning energy production at the ankle joint compared to the transition and preferred speeds.

The earlier initiation of the plantar flexion in the two higher speeds (47% of the stance phase) compared to the slow and preferred ones (72% of the stance phase) was associated with a rapid increase in the EMG-activity of the Sol and GM muscles and with a knee-to-ankle joint energy transfer via the biarticular gastrocnemii muscles (Appendix A). Interestingly, the dorsiflexion and plantar flexion duration intersected at the transition speed, indicating a different coordination at the ankle joint as well as between ankle and knee joints in the two higher walking speeds that promote the increase of the ATF work. The earlier initiation of the plantar flexion (relative to the stance phase) combined with a rapid increase in the EMG-activity of the Sol and GM resulted in an increase of the AT force in the two higher walking speeds during the first part of the plantar flexion, whereas in the slow and preferred speed the plantar flexion was associated with a continuous and rapid decline of AT force. The result was an invariant (*p* = 0.295) average AT force in all speeds during the plantar flexion phase (slow: 1035 ± 148 N, preferred: 1175 ± 143 N, transition: 1016 ± 102 N, maximum: 985 ± 128 N). The similar average AT force with the increased plantar flexion RoM resulted to a greater positive ATF work at the maximum walking speed. The lower negative ATF work at the transition speed and the higher positive ATF work at the maximum speed caused the increase of net ATF work despite the decrease of maximum AT force in the two higher speeds. This earlier initiation of the plantar flexion occurred at a low AT operating strain (Figure 6). At the beginning of the plantar flexion, the AT strain was 2.6 ± 0.2% in the transition and 3.3 ± 0.3% in the maximum walking speed, and both were clearly lower in comparison during slow 3.9 ± 0.3% and preferred 4.1 ± 0.3% walking speeds. The low AT strain values in the two higher speeds indicate that the initiation of the plantar flexion and, thus, the production of mechanical work at the ankle joint occurred when the AT operates closer to or even within the toe region of the tendon force-elongation relationship, where the tendon may be elongated with rather low increments of muscle force. The energy transfer from the knee-to-ankle joint via the biarticular gastrocnemii muscles and the rapid increase in Sol and GM muscle activation at a low level of muscle forces (26 ± 0.4% of the MVC) facilitated the muscles’ force generation and storage of the elastic strain energy on the AT even during the plantar flexion. The simultaneous plantar flexion and AT elongation during the transition and maximum speeds show, at least for the monoarticular Sol muscle, a transfer of contractile energy production to the AT. Furthermore, the knee-to-ankle joint energy transfer via the gastrocnemii muscles indicates energy transfer from the proximal quadriceps muscles to the AT.

The biarticular net work at the ankle joint increased as a function of walking speed and was greater than the net mechanical work of the gastrocnemii MTU, particularly at the two higher speeds. In addition, the monoarticular Sol muscle contractile energy production was greater in the transition and maximum walking speed compared to the preferred one. Although, the contractile energy production of the monoarticular Sol and the participation of the biarticular mechanisms of the gastrocnemii muscles may increase the net mechanical work done at the ankle joint during the two higher speeds, and thus walking performance is metabolically costly. Contractile work during shortening increases the active muscle volume and thus the metabolic costs. The enormous increased EMG activity of the investigated muscles in the transition and maximum walking speeds predict a higher active muscle volume. In the biarticular knee-to-ankle joint energy-transfer mechanism, the more proximal quadriceps muscles are involved, and part of their contractile work can be transferred to the ankle joint [8,51]. Due to the greater muscle volume of the quadriceps compared to the distal Sol, GM, and GL muscles [52,53], they can produce more muscular power and work. However, due to their longer fibers, a unit of force generation is metabolically more expensive compared to the shorter Sol, GM, and GL muscles [54,55]. Therefore, the knee-to-ankle joint energy transfer at the two higher speeds might increase the metabolic costs of walking. Studies that measured the metabolic cost of walking show that the energy cost increased at speeds higher than the preferred and at speeds above 2.2 m.s${}^{-1}$, e.g., our maximum speed, (2.6 m.s${}^{-1}$), it is even higher than during running [23].

The quantification of AT force in vivo is challenging, and there are some limitations associated with the approach used in the present study that need to be considered when interpreting the results. The anatomical structure of the AT comprises bundles of fascicles originating from Sol, GM, and GL muscles [56]. This structure allows a certain degree of inter-fascicle sliding, which may result in different regional displacement gradients within the AT during loading [57]. Contraction dynamics of the Sol, GM, and GL muscles may affect the inter-fascicle sliding within the AT during walking, which would affect the accuracy of the AT force quantification. On the other hand, there is also evidence that the myotendinous structures of the Sol, GM, and GL do not work independently but are mechanically connected [58,59]. Myotendinous force transmission through contiguous extramuscular connective tissue structures influences the mechanical linkage of adjacent MTUs during muscle contraction [60,61] and promotes a synchronous movement of connective tissues between the Sol, GM, and GL muscles [58,59]. Force transmission mechanisms through the connective tissue network of the MTUs may redistribute the forces within the AT in order to minimize peak stresses [62] and thus, heterogeneous regional displacement gradients. Similar AT force–elongation curves have been reported when using either the myotendinous junction of the GM or GL as measured points [63,64], and were independent of the ankle angle, i.e., rest length of the AT [63,65]. Consequently, although a certain degree of heterogeneous displacement within the AT length is possible during walking (which may, to some extent, influence the accuracy of the AT force quantification) it is unlikely to change the main findings and conclusions.

In this study, the AT force contribution of each individual muscle was assumed to be proportional to its relative PCSA. However, the individual muscle force also depends on the respective force–length–velocity potential (i.e., fraction of maximum force according to the force–length and force–velocity curves) and muscle activation. During human walking, the force–length–velocity potential of the Sol and GM was found to be between 0.6 and 0.8 [66,67]. We performed a sensitivity analysis by varying the force-length–velocity potential of the Sol and gastrocnemii separately in 0.1 intervals between 0.6 and 0.8. Since a similar fascicle shortening behavior of the GM and GL has been found during the stance phase of walking [68], we assumed a similar force–length–velocity potential for the two muscles. To estimate muscular activation, we used the average normalized EMG-activity for the Sol and GM muscles we measured during walking. Some studies [68,69] reported distinct activation patterns of the GM and GL during walking. Hamard et al. [68] reported a ratio of the GM EMG-activity to the sum of GM and GL EMG-activity between 0.44 and 0.68 across participants. Since we did not measure the EMG-activity of the GL, an activity-ratio of GM from 0.4 to 0.7 in 0.1 increments was used for the sensitivity analysis. In total, 36 different combinations were considered. The new distribution coefficients ($k_{i}$) used for the sensitivity analysis were calculated with Equation (3):(3)$$k_{i}=\frac{\lambda_{i}^{l,v}.\alpha_{i}.r_{i}^{PCSA}}{\sum_{i}^{3}\lambda_{i}^{l,v}.\alpha_{i}.r_{i}^{PCSA}}$$
with *i*: Sol, GM, GL; $\lambda_{i}^{l,v}$: force–length–velocity potential of the muscle *i*; $\alpha_{i}$: activation of muscle *i*; $r_{i}^{PCSA}$: relative PCSA of muscle *i*.

The Appendix A show the results of the sensitivity analysis on the net work of the Sol and gastrocnemii at the ankle joint as well as the net work of the gastrocnemii at the knee joint, which represents the contribution of the biarticular mechanisms to the increase of net ankle work. Different force potentials and ratios of the GM EMG activity affect the absolute values in all three parameters. Nevertheless, in all combinations, the contribution of the Sol to the net mechanical work at the ankle joint was higher than the gastrocnemii. Both the energy production of the monoarticular Sol and the contribution of the biarticular mechanisms (i.e., increased net ankle work) increased with walking speed. Thus, the sensitivity analysis clearly demonstrates that the main findings and conclusions generally hold true even under the assumption of varying force–length–velocity potentials and activation of the individual muscles.

In conclusion, an increase in the net mechanical work at the ankle joint from the Sol and gastrocnemii muscles is needed to increase walking speed. The net mechanical work at the ankle joint can be affected by the contractile energy production of the three muscles and by the biarticular mechanisms of the gastrocnemii muscles. In the current study, we focused on the contribution of monoarticular and biarticular muscle mechanisms for the needed greater net ATF work from slow to maximum walking speeds. Our findings show that both the contractile energy production of the monoarticular Sol muscle and the biarticular mechanisms of the gastrocnemii muscles contributed to the speed-related increased net mechanical work at the ankle joint. Although the contribution of the contractile work of the Sol was greater, biarticular mechanisms of the gastrocnemii accounted for a relevant part of the increased net ankle joint mechanical work with speed. An earlier plantar flexion initiated by a rapidly increased activation of the Sol and GM muscles increased the net ATF mechanical work 1.7 and 2.4-fold in the transition and maximum walking speeds, respectively, compared to the preferred one, despite a decrease in maximum AT force.

## Figures and Tables

**Figure 1 biology-12-00872-f001:**
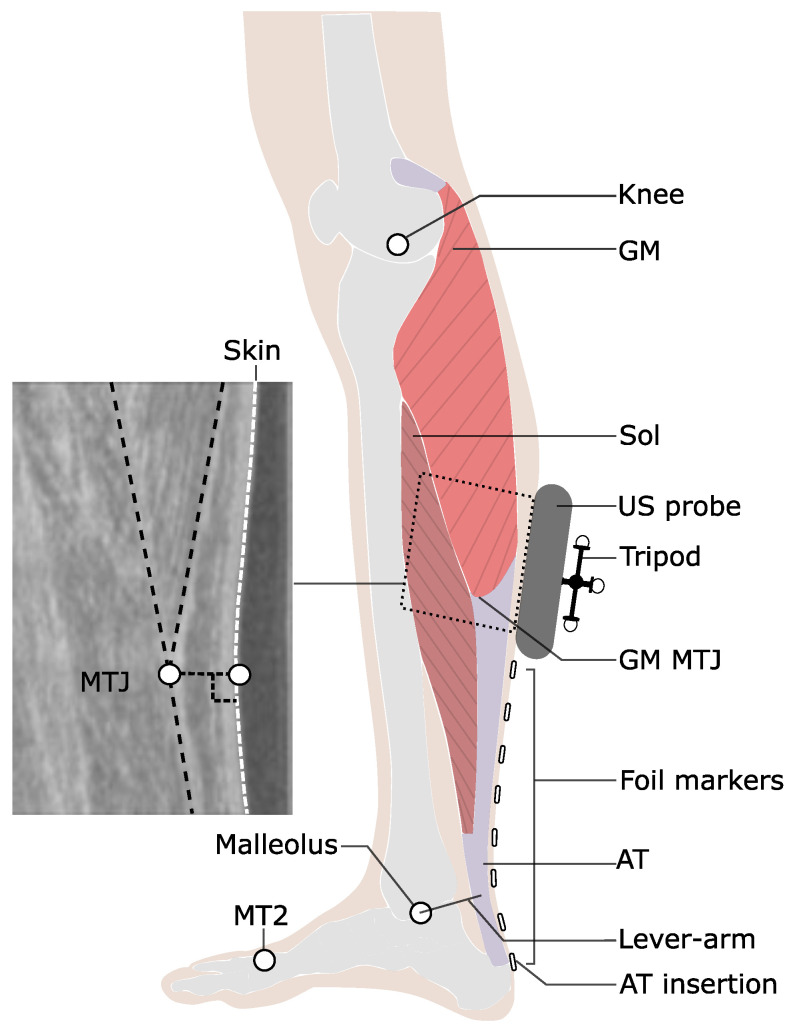
Experimental setup for the determination of Achilles tendon (AT) length during locomotion. Reflective foil markers were used to reconstruct the curved shape of the AT. An ultrasound (US) probe was used to detect the movements of gastrocnemius medialis myotendinous junction (GM-MTJ), which was then projected to the skin surface (white dashed line). The tripod markers were used to transfer the detected positions of the GM-MTJ to the global coordinate system (same as the motion capture system). The AT length was calculated as the sum of Euclidian distances from AT insertion (notch of calcaneus bone) between every two consecutive foil markers until the projected position of GM-MTJ to the skin surface. The AT lever arm was defined as the perpendicular distance between the midline of AT curved path and the ankle joint center. MT2: the tip of the second metatarsal; Sol: soleus muscle; Knee: a reflective marker on the epicondyle of the femur.

**Figure 2 biology-12-00872-f002:**
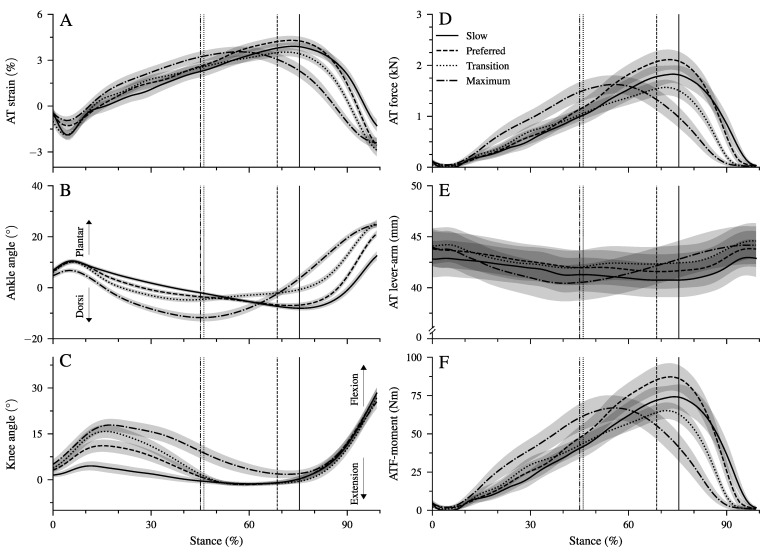
Achilles tendon (AT) strain (**A**), ankle and knee joint angles (**B**,**C**), AT force (**D**), AT lever arm (**E**), and moment generated from the AT force at the ankle joint (ATF moment, (**F**) during the stance phase of walking at slow (0.7 m.s${}^{-1}$, Slow), preferred (1.4 m.s${}^{-1}$, Preferred), transition (2.0 m.s${}^{-1}$, transition), and maximum speed (2.6 ± 0.3 m.s${}^{-1}$, Max). The vertical solid, dashed, dotted, and dashed–dotted lines separate the dorsiflexion and plantar flexion of the ankle during walking at slow, preferred, transition, and maximum speed, respectively. The curves and shaded areas represent mean ± standard errors (average of fifteen participants with nine gait cycles).

**Figure 3 biology-12-00872-f003:**
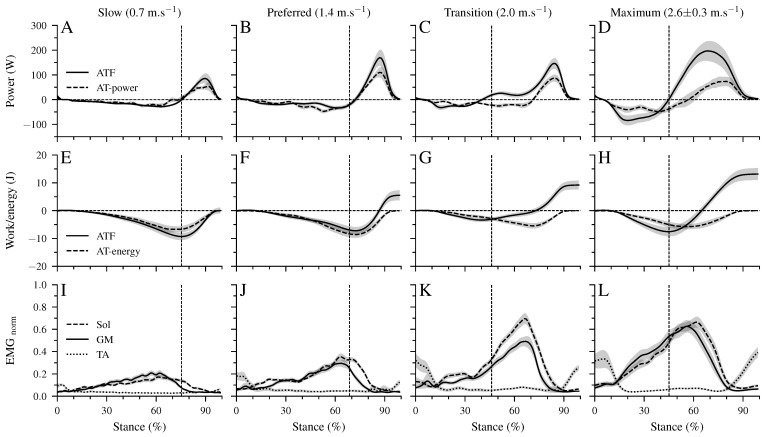
Mechanical power (**A**–**D**) and work (**E**–**H**) of the AT force at the ankle joint (ATF power/work), as well as Achilles tendon power (AT power) and elastic strain energy storage and recoil (AT energy). Negative values in the mechanical power indicate energy absorption at the ankle joint during dorsiflexion and AT elastic strain energy storage. Positive power values indicate energy production at the ankle joint during the plantar flexion and elastic AT strain energy recoil. Panel (**I**–**L**) show the electromyographic activity of the soleus (Sol), gastrocnemius medialis (GM), and tibialis anterior (TA) muscles normalized to a maximum voluntary contraction (EMGnorm). All investigated parameters are presented for the stance phase of walking at slow (0.7 m.s${}^{-1}$), preferred (1.4 m.s${}^{-1}$), transition (2.0 m.s${}^{-1}$), and maximum speed (2.6 ± 0.3 m.s${}^{-1}$) as mean ± standard error (average of fifteen participants with nine gait cycles). The vertical dashed line shows the separation between dorsiflexion and plantar flexion.

**Figure 4 biology-12-00872-f004:**
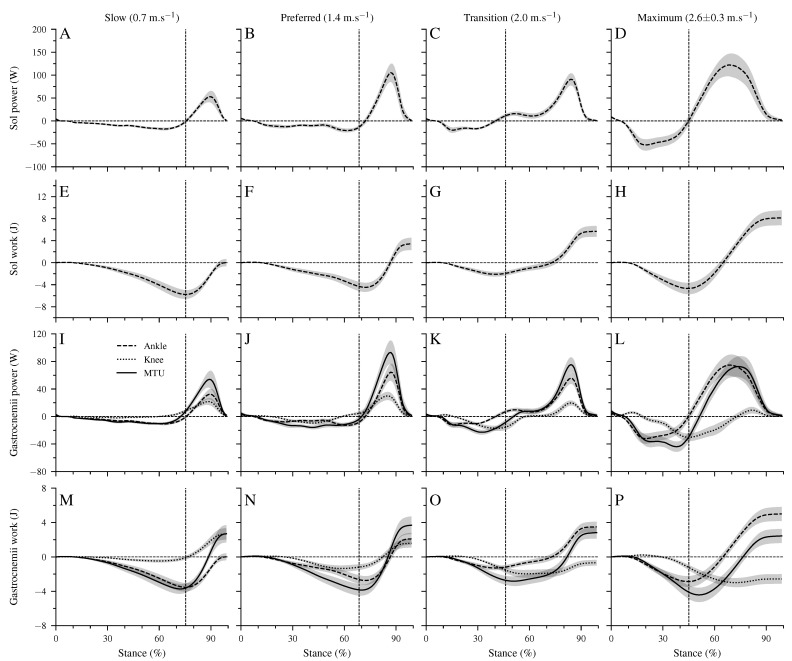
Mechanical power (**A**–**D**) and work (**E**–**H**) of the soleus (Sol) muscle at the ankle joint, mechanical power (**I**–**L**) and work (**M**–**P**) of the gastrocnemii muscles at the ankle and knee joints, and the mechanical power/work of the gastrocnemii muscle-tendon unit (MTU). Negative values in the mechanical power indicate energy absorption at the ankle joint during dorsiflexion, energy absorption at the knee joint during knee extension, and energy absorption of the MTU during lengthening. Positive power values indicate energy production during ankle plantar flexion, knee flexion, and MTU shortening. All investigated parameters are presented for the stance phase of walking at slow (0.7 m.s${}^{-1}$), preferred (1.4 m.s${}^{-1}$), transition (2.0 m.s${}^{-1}$), and maximum speed (2.6 ± 0.3 m.s${}^{-1}$) as mean ± standard error (average of fifteen participants with nine gait cycles). The vertical dashed line shows the separation between dorsiflexion and plantar flexion.

**Figure 5 biology-12-00872-f005:**
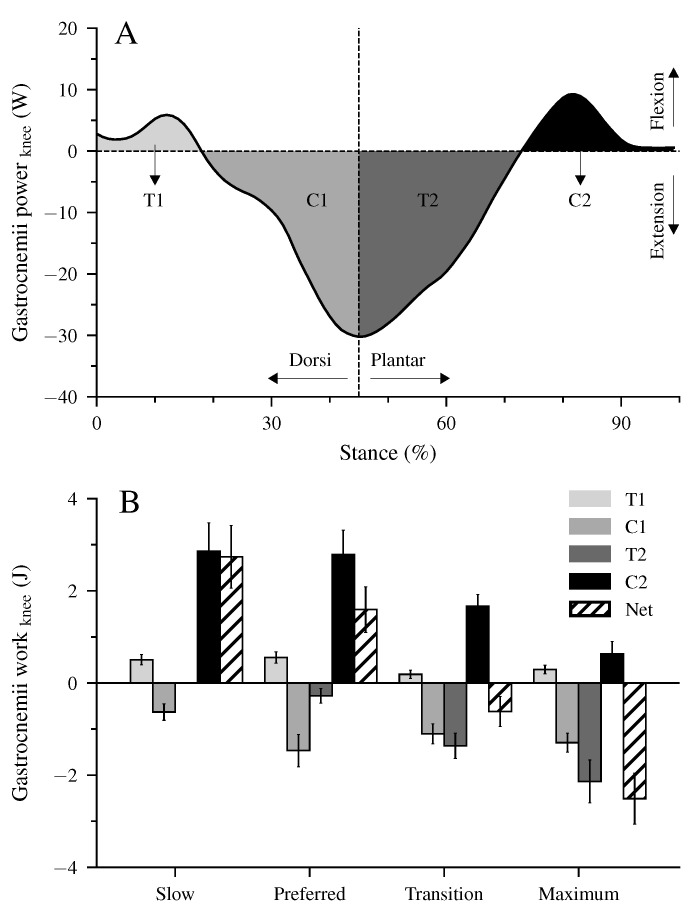
Mechanical power (**A**) and work (**B**) of the biarticular gastrocnemii muscles at the knee joint illustrate the transfer (T) and joint-coupling mechanisms (C). T1: ankle-to-knee joint energy transfer, C1: energy absorption at the knee joint during dorsiflexion and knee extension, T2: knee-to-ankle joint energy transfer, C2: energy production at the knee joint during plantar flexion and knee flexion, Net: net work at the knee joint. The values are mean ± standard error (average of fifteen participants with nine gait cycles).

**Figure 6 biology-12-00872-f006:**
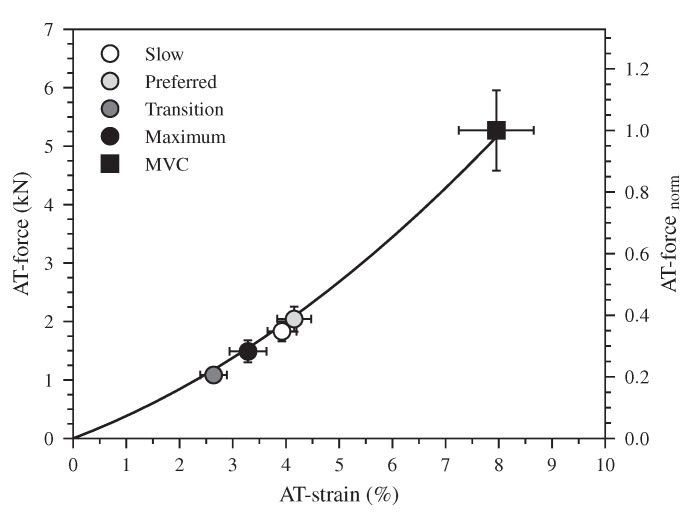
The force-strain relationship of the Achilles tendon (AT) during a maximum voluntary contraction (MVC). The markers show the operating AT force and strain values at the beginning of the plantar flexion during walking at slow (0.7 m.s${}^{-1}$), preferred (1.4 m.s${}^{-1}$), transition (2.0 m.s${}^{-1}$) and maximum speed (2.6 ± 0.3 m.s${}^{-1}$, Max) as means ± standard error (average of fifteen participants with nine gait cycles).

**Table 1 biology-12-00872-t001:** Spatiotemporal gait parameters during the stance phase of walking at slow (0.7 m.s${}^{-1}$), preferred (1.4 m.s${}^{-1}$), transition (2.0 m.s${}^{-1}$) and maximum speed (2.6 ± 0.3 m.s${}^{-1}$, Max). All values are presented as mean ± standard error (average of fifteen participants with nine gait cycles). * Statistically significant effect of speed (*p* < 0.05). Each row sharing the same letter does not differ significantly (*p* > 0.05, post hoc analysis).

	Slow $\left(0.7\;m.s^{-1}\right)$	Preferred $\left(1.4\;m.s^{-1}\right)$	Transition $\left(2.0\;m.s^{-1}\right)$	Maximum $\left(2.6\pm 0.3\;m.s^{-1}\right.$)
Stance time (ms) *	863 ± 27 $\underline{a}$	$601\pm 11\;\underline{b}$	477 ± 09 $\underline{c}$	363 ± 11 $\underline{d}$
Dorsiflexion time (ms) *	$652\pm 15\;\underline{a}$	$416\pm 20\;\underline{b}$	$229\pm 13\;\underline{c}$	$167\pm 06\;\underline{d}$
Plantar flexion time (ms) *	$211\pm 14\;\underline{ab}$	$185\pm 15\;\underline{a}$	$249\pm 19\;\underline{b}$	$197\pm 09\;\underline{a}$
Swing time (ms) *	$490\pm 11\;\underline{a}$	$411\pm 08\;\underline{b}$	$355\pm 06\;\underline{c}$	$316\pm 5\;\underline{d}$
Step length (m) *	$0.47\pm 0.01\;\underline{a}$	$0.7\pm 0.01\;\underline{b}$	$0.83\pm 0.01\;\underline{c}$	$0.89\pm 0.02\;\underline{d}$
Cadence (steps/s) *	$1.49\pm 0.03\;\underline{a}$	$1.98\pm 0.03\;\underline{b}$	$2.41\pm 0.04\;\underline{c}$	$2.95\pm 0.06\;\underline{d}$
Duty-factor *	$0.64\pm 0\;\underline{a}$	$0.59\pm 0\;\underline{b}$	$0.57\pm 0\;\underline{c}$	$0.53\pm 0.01\;\underline{d}$

**Table 2 biology-12-00872-t002:** Peak values of the Achilles tendon (AT) strain and force, average AT lever arm, maximum moment of the AT force at the ankle joint (ATF moment) and average range of motion (RoM) at the ankle joint during the dorsiflexion and plantar flexion phase at slow (0.7 m.s${}^{-1}$), preferred (1.4 m.s${}^{-1}$), transition (2.0 m.s${}^{-1}$) and maximum walking speed (2.6 ± 0.3 m.s${}^{-1}$, Max). All values are presented as mean ± standard error (average of fifteen participants with nine gait cycles). * Statistically significant effect of speed (*p* < 0.05). Each row that shares the same letter does not differ significantly (*p* > 0.05, post hoc analysis).

	Slow $\left(0.7\;m.s^{-1}\right)$	Preferred $\left(1.4\;m.s^{-1}\right)$	Transition $\left(2.0\;m.s^{-1}\right)$	Maximum $\left(2.6\pm 0.3\;m.s^{-1}\right.$)
AT strain (%) *	4.1± 0.3 $\underline{ab}$	$4.4\pm 0.3\;\underline{a}$	$3.6\pm 0.3\;\underline{b}$	$3.8\pm 0.4\;\underline{ab}$
AT force (N) *	$1930\pm 190\;\underline{ab}$	$2174\pm 202\;\underline{a}$	$1616\pm 147\;\underline{b}$	$1793\pm 2.3\;\underline{b}$
AT lever arm (mm) *	$41\pm 1\;\underline{a}$	$42\pm 1\;\underline{ab}$	$43\pm 1\;\underline{b}$	$42\pm 1\;\underline{a}$
ATF moment (Nm) *	$78\pm 08\;\underline{ab}$	$90\pm 09\;\underline{a}$	$67\pm 05\;\underline{b}$	$74\pm 09\;\underline{ab}$
Ankle ROM dorsiflexion $\left({}^{\circ }\right)$ *	$15\pm 1\;\underline{ab}$	$14\pm 1\;\underline{a}$	$11\pm 1\;\underline{c}$	$17\pm 2\;\underline{b}$
Ankle ROM plantar flexion $\left({}^{\circ }\right)$ *	$21\pm 1\underline{b}$	$29\pm 1\;\underline{a}$	$30\pm 1\;\underline{a}$	$37\pm 1\;\underline{c}$

**Table 3 biology-12-00872-t003:** Mechanical work of the Achilles tendon force (ATF work) during dorsiflexion, plantar flexion, and net ATF work at the ankle joint. Achilles tendon energy storage and recoil (AT energy) and normalized maximum electromyographic activity of the soleus (Sol EMG norm), gastrocnemius medialis (GM EMG norm), and tibialis anterior (TA EMG norm) muscles at slow (0.7 m.s${}^{-1}$), preferred (1.4 m.s${}^{-1}$), transition (2.0 m.s${}^{-1}$), and maximum walking speed (2.6 ± 0.3 m.s${}^{-1}$, Max). All values are presented as mean ± standard error (average of fifteen participants with nine gait cycles). * Statistically significant effect of speed (*p* < 0.05). Each row that shares the same letter does not differ significantly (*p* > 0.05, post hoc analysis).

	Slow $\left(0.7\;m.s^{-1}\right)$	Preferred $\left(1.4\;m.s^{-1}\right)$	Transition $\left(2.0\;m.s^{-1}\right)$	Maximum $\left(2.6\pm 0.3\;m.s^{-1}\right.$)
ATF work dorsiflexion (J) *	−9.6 ± 1.3 $\underline{a}$	−8.0 ± 1.2 $\underline{a}$	−4.0 ± 0.7 $\underline{b}$	$-8.1\pm 1.7\;\underline{a}$
ATF work plantar flexion (J) *	$9.6\pm 2.0\;\underline{a}$	$13.5\pm 2.5\;\underline{a}$	$13.17\pm 2.0\;\underline{a}$	$21.3\pm 3.2\;\underline{b}$
ATF work net (J) *	$0.0\pm 1.1\;\underline{a}$	$5.5\pm 1.8\;\underline{b}$	$9.19\pm 1.6\;\underline{c}$	$13.2\pm 2.1\;\underline{d}$
AT energy (stored/recoiled, J) *	$7.2\pm 1.1\;\underline{ab}$	$8.8\pm 1.1\;\underline{a}$	$5.66\pm 0.9\;\underline{b}$	$6.7\pm 1.1\;\underline{ab}$
Sol EMG${}_{\mathrm{norm}\;}$ *	$0.20\pm 0.02\;\underline{a}$	$0.39\pm 0.02\;\underline{b}$	$0.73\pm 0.04\;\underline{c}$	$0.77\pm 0.04\;\underline{c}$
GM EMG${}_{\mathrm{norm}\;}$ *	$0.25\pm 0.02\;\underline{a}$	$0.35\pm 0.03\;\underline{a}$	$0.61\pm 0.06\;\underline{b}$	$0.71\pm 0.06\;\underline{c}$
TA ${\mathrm{EMG}}_{\mathrm{norm}\;}$ *	$0.13\pm 0.02\;\underline{a}$	$0.22\pm 0.04\;\underline{a}$	$0.38\pm 0.06\;\underline{b}$	$0.51\pm 0.06\;\underline{c}$

**Table 4 biology-12-00872-t004:** Negative, positive, and net work of the soleus (Sol work) muscle at the ankle joint, the gastrocnemii muscles (gastro work) at the ankle and knee joints, and the gastrocnemii muscle–tendon unit (Gastro MTU work) at slow (0.7 m.s${}^{-1}$), preferred (1.4 m.s${}^{-1}$), transition (2.0 m.s${}^{-1}$), and maximum walking speed (2.6 ± 0.3 m.s${}^{-1}$, Max). All values are presented as mean ± standard error (average of fifteen participants with nine gait cycles). * Statistically significant effect of speed (*p* < 0.05). Each row sharing the same letter does not differ significantly (*p* > 0.05, post hoc analysis).

		Slow $\left(0.7\;m.s^{-1}\right)$	Preferred $\left(1.4\;m.s^{-1}\right)$	Transition $\left(2.0\;m.s^{-1}\right)$	Maximum $\left(2.6\pm 0.3\;m.s^{-1}\right.$)
Sol work Ankle	Negative (J) *	$-5.9\pm 0.8\;\underline{a}$	$-4.9\pm 0.7\;\underline{a}$	$-2.5\pm 0.4\;\underline{b}$	$-5.0\pm 1.0\;\underline{a}$
	Positive (J) *	$5.9\pm 1.2\;\underline{a}$	$8.4\pm 1.6\;\underline{a}$	$8.1\pm 1.3\;\underline{a}$	$13.1\pm 1.0\;\underline{b}$
	Net(J) *	$0\pm 0.7\;\underline{a}$	$3.4\pm 1.1\;\underline{b}$	$5.7\pm 1.0\;\underline{c}$	$8.1\pm 1.3\;\underline{d}$
Gastro work Ankle	Negative (J) *	$-3.7\pm 0.5\;\underline{a}$	$-3.1\pm 0.5\;\underline{a}$	$-1.5\pm 0.3\;\underline{b}$	$-3.1\pm 0.7\;\underline{a}$
	Positive (J) *	$3.8\pm 0.8\;\underline{a}$	$5.2\pm 1.0\;\underline{a}$	$5.0\pm 0.8\;\underline{a}$	$8.1\pm 1.2\;\underline{b}$
	Net(J) *	$0.1\pm 0.4\;\underline{b}$	$2.1\pm 0.7\;\underline{a}$	$3.5\pm 0.6\;\underline{c}$	$5.0\pm 0.8\;\underline{d}$
Gastro work Knee	Negative (J) *	$-0.6\pm 0.1\;\underline{b}$	$-1.7\pm 0.4\;\underline{a}$	$-2.5\pm 0.5\;\underline{a}$	$-3.4\pm 0.6\;\underline{c}$
	Positive (J) *	$3.4\pm 0.7\;\underline{a}$	$3.3\pm 0.6\;\underline{a}$	$1.8\pm 0.3\;\underline{b}$	$0.9\pm 0.3\;\underline{c}$
	Net(J) *	$2.7\pm 0.7\;\underline{a}$	$1.6\pm 0.5\;\underline{a}$	$-0.6\pm 0.3\;\underline{b}$	$-2.5\pm 0.5\;\underline{c}$
Gastro MTU work	Negative (J)	−3.9±0.6	−4.2±0.7	−3.0±0.6	−4.6±0.9
	Positive (J)	6.6±1.3	7.9±1.4	5.9±0.9	7.1±1.1
	Net(J)	2.7±1.0	3.7±1.0	2.8±0.7	2.4±0.8

## Data Availability

https://doi.org/10.6084/m9.figshare.21075907.v3.

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
