# Peer review of "Contractile Work of the Soleus and Biarticular Mechanisms of the Gastrocnemii Muscles Increase the Net Ankle Mechanical Work at High Walking Speeds"

_biology, 2023, doi:10.3390/biology12060872_

Round 1
Reviewer 1 Report
The goal of this study was to investigate the contribution of one-joint soleus and two-joint gastrocnemius muscles to mechanical power and work at walking speeds 0.7 m/s, 1.4 m/s (preferred walking speed), 2.0 m/s (walk-run transition speed) and 2.6 m/s (maximum walking speed) in humans. In addition, the authors investigated energy transfer between the knee and ankle by the two-joint gastrocnemius muscle. The authors used inverse dynamic analysis and measurements of Achilles tendon length changes using an ultrasound probe to compute power and work at the ankle and knee as well as power and work produced/done by the one-joint soleus at the ankle and power and work of the two-joint gastrocnemius at the ankle and knee. The authors demonstrated that the maximum Achilles tendon force decreased by 21% at the transition and maximum walking speeds, however the Achilles tendon force and its work increased with increasing walking speeds. An increased activity of soleus and gastrocnemius and the contribution of energy transfer from the knee to the ankle at the transition and maximal speeds increase the Achilles tendon work by 1.7 and 2.4 times at the two highest walking speeds.
The study provides new important results on the function of the ankle extensors at different walking speeds and the energy transfer by the two-joint gastrocnemius from the knee to the ankle during human walking, the first such demonstration. The manuscript is well written. Introduction is complete and provides a clear rationale for the study. The methods are sound and described in detail and statistical methods appear appropriate. The results are well illustrated – the figures and tables are informative and contain information sufficient to evaluate authors’ conclusions.
I reviewed an earlier version of this manuscript submitted to another journal. I made a number of comments and suggestions, and the authors did a good job to address them and edit the manuscript accordingly. Given that all my previous comments are properly addressed, I do not have any additional comments.
Author Response
We greatly appreciate your expertise and the thoroughness you assessed our manuscript. We want to express our gratitude for the positive aspects of your review. Your recognition of the methodology employed and the clarity of the results and conclusions has served as a tremendous encouragement for us. Your kind words have reaffirmed our belief in the significance and relevance of this research.
Reviewer 2 Report
The authors previously proposed and validated an elegant method for measuring the work performed by the plantar flexor muscles during real locomotion using an ultrasound sensor system coupled with EMG measurements. They reused this original method in the present study during walking to measure the work provided by each of the plantar flexor muscles, taking into account their viscoelastic properties. This important work for our fundamental understanding of human locomotion reveals the existence of energy transfer between the knee and ankle joints, mediated by the biarticular gastrocnemius muscle, as already seen in running dogs. At high walking speeds, the powerful quadriceps muscles appear to transfer their energy to ankle through the biarticular gastrocnemius muscles. Congratulations to the authors. The present study represents an advancement in our knowledge of human locomotion, confirming the mechanism that had already been hypothesized by van Ingen Schenau 30 years earlier. In addition to these proximal-to-distal energy transfers, the plantar flexor muscles also increase their work beyond the transition speed, providing an explanation for (i) the increase in energy cost beyond the preferred walking speed and (ii) the causes of the gait transition from walking to running.
A suggestion is made to enhance the contribution of this study to our knowledge:
Include a diagram that illustrates the mechanisms of proximal-to-distal energy transfer, particularly at high walking speeds, highlighting the contribution of the large proximal muscle masses represented by the quadriceps to the ankle joint work. This diagram could partly explain the increased energy cost of walking beyond the preferred walking speed.
Minor comment: Provide clarification in Figure 3, please indicate the curves corresponding to Achilles tendon power (AT-power) and elastic strain energy storage and recoil (AT-energy), respectively.
Author Response
Comment: The authors previously proposed and validated an elegant method for measuring the work performed by the plantar flexor muscles during real locomotion using an ultrasound sensor system coupled with EMG measurements. They reused this original method in the present study during walking to measure the work provided by each of the plantar flexor muscles, taking into account their viscoelastic properties. This important work for our fundamental understanding of human locomotion reveals the existence of energy transfer between the knee and ankle joints, mediated by the biarticular gastrocnemius muscle, as already seen in running dogs. At high walking speeds, the powerful quadriceps muscles appear to transfer their energy to ankle through the biarticular gastrocnemius muscles. Congratulations to the authors. The present study represents an advancement in our knowledge of human locomotion, confirming the mechanism that had already been hypothesized by van Ingen Schenau 30 years earlier. In addition to these proximal-to-distal energy transfers, the plantar flexor muscles also increase their work beyond the transition speed, providing an explanation for (i) the increase in energy cost beyond the preferred walking speed and (ii) the causes of the gait transition from walking to running.
Answer: Many thanks for your time and effort invested in reviewing our manuscript.
Comment: A suggestion is made to enhance the contribution of this study to our knowledge:
Include a diagram that illustrates the mechanisms of proximal-to-distal energy transfer, particularly at high walking speeds, highlighting the contribution of the large proximal muscle masses represented by the quadriceps to the ankle joint work. This diagram could partly explain the increased energy cost of walking beyond the preferred walking speed.
Answer: Thank you for your valuable suggestion. We have provided an animation video illustrating the energy transfer from knee to ankle in the supplementary files.
The video clip illustrates a graphical animation of the right leg during the maximal walking speed of an individual. The vastus lateralis and the gastrocnemii muscles are colored in red, and the soleus in the blue spectrum. Lighter red and blue colors show lower muscle activity, and darker red and blue colors show higher muscle activity. From 47% to 72% of the stance phase, the knee joint extends, and the ankle joint is plantarflexing. This simultaneous knee extension and ankle plantar flexion initiate the energy transfer from the knee extensor to the ankle plantar flexor muscles. Here you can find the video in supplementary materials.
Comment: Provide clarification in Figure 3. Please indicate the curves corresponding to Achilles tendon power (AT-power) and elastic strain energy storage and recoil (AT-energy), respectively.
Answer: Thank you for the comment. The legend in Figure 3 is now corrected.